# Intra-Pulse Frequency Coding Design for a High-Resolution Radar against Smart Noise Jamming

**Rongyan Xi [1], Dingyou Ma [2], Xiang Liu [3], Lei Wang [1],\* and Yimin Liu [1]**

[1] Department of Electronic Engineering, Tsinghua University, Beijing 100084, China; xiry17@mails.tsinghua.edu.cn (R.X.); yiminliu@tsinghua.edu.cn (Y.L.)

[2] Key Laboratory of Universal Wireless Communications, Ministry of Education, School of Information and Communication Engineering, Beijing University of Posts and Telecommunications, Beijing 100876, China; dingyouma@bupt.edu.cn

[3] Institute of Electronic Engineering, China Academy of Engineering Physics, Mianyang 621900, China; x-liu12@foxmail.com

\* Correspondence: leiwangqh@tsinghua.edu.cn

**Abstract:** Smart noise jamming forms active jamming by intercepting, modulating, and forwarding radar signals into the radar receiver, which seriously affects the radar range recovery performance. In this paper, we propose a novel waveform design approach and an efficient range recovery method for high-resolution radar in the jamming scenario. Firstly, we propose an intra-pulse frequency-coded frequency-modulated continuous waveform (IPFC-FMCW), which contains multiple FMCW chips with different widths and frequencies, to combat the smart noise jamming. After the jamming suppression, the proposed waveform has a low sidelobe level, which is different from traditional FMCW signals for which the observations are periodically missing, resulting in high sidelobe levels. Then, to improve the range recovery performance of the waveform after jamming suppression, we optimize the range profile by designing the transmit waveform and then solve it by a simulated annealing algorithm. Next, based on the designed waveform, we derive the echo model after jamming suppression and propose a gridless compressed sensing (CS) method to recover the range of the targets. Compared with the existing waveforms and methods, the proposed waveform and the processing method achieve better range recovery performance in the jamming scenario. Numerical simulations are utilized to demonstrate the range recovery effectiveness of the proposed waveform and method in smart noise jamming.

**Keywords:** high-resolution radar; frequency coding; smart noise jamming; waveform design

## 1. Introduction

HIGH-resolution range recovery of a high-resolution radar has many potential applications in multi-dimensional imaging and target recognition [1–5]. To obtain a high range resolution, signals with large bandwidths are generally transmitted. The realization of a large bandwidth can be divided into two categories. The first type transmits real wideband signals, while the other type radiates a series of narrowband waveforms to achieve the range resolution of a wideband waveform by bandwidth synthesizing approaches. Although the bandwidth synthesizing techniques decrease the instantaneous bandwidth of radar systems, the requirement of a long observation time limits the application of these methods in the probing of fast moving targets [6]. In contrast, a high-resolution radar that directly transmits wideband signals only needs to process within the duration of a single pulse, which is advantageous for the detection of non-cooperative targets moving at high speed, and it has attracted extensive research attention [7,8].

Early studies mainly focused on improving the quality of range recovery in the absence of jamming. Many studies modeled range recovery as a sparse recovery problem and used compressed sensing (CS) methods to solve it [9–12]. However, with the rapid development

of digital radio frequency memory (DRFM) technology, the coherent jamming significantly degrades the quality of range recovery [13,14]. In the jamming scene, the jammer transmits intercepted signals to the radar, which has the advantages of easy implementation and fast response [15]. With the development of smart noise jamming, the jammer modulates the intercepted signal and generates smart noise jamming [16,17], which has both blanket noise jamming and deception jamming effects. Due to the influence of jamming, observations are periodically polluted, which seriously affects the performance of range recovery. Consequently, jamming suppression is one of the most pressing issues for range recovery of a high-resolution radar.

In recent years, jamming suppression has become one of the research hotspots in the field of electronic counter-countermeasures, and many methods of jamming suppression have been proposed, including signal processing algorithms and waveform design approaches. Research [18] reconstructed the jamming signal and eliminated the jamming signal by estimating the jamming parameters, which requires an accurate estimation of the jamming parameters. According to the discontinuity of interrupted-sampling repeater jamming (ISRJ) in the time-frequency domain, some filtering methods were proposed in [19–22], which effectively estimate the jamming parameters and filter the ISRJ in the frequency domain. However, these methods are no longer applicable to smart noise jamming because the target echo and the jamming signal coincide in the frequency domain. In addition, the direct elimination of jamming slices causes a periodic loss of observation data, and the problem caused by grating lobes is severe.

In addition to signal processing algorithms, waveform design is also an effective way to solve jamming problems. Some researchers focus on the design of orthogonal coded waveforms to mismatch the ISRJ with unintercepted radar signal slices [23–25]. However, generating such a phase-coded waveform is difficult when the wideband signal is required. The frequency-modulated continuous waveform (FMCW) has attracted much attention because it is easy to generate wideband signals using low-cost hardware. However, with the development of interrupted sampling technology, the jamming appears periodically, and the observations are periodically missing due to the jamming suppression, which causes a serious grating lobe problem. The carrier-frequency agile FMCW signal refers to the FMCW signals with different carrier frequencies in each period, which has random characteristics and improves the radar anti-jamming performance. An inter-pulse carrier-frequency agile FMCW signal for a dual-function radar communication system is proposed in [26], while the waveform cannot cope with the rapidly changing jamming within a pulse. In order to cope with the jamming within a pulse, inspired by frequency-agile radar, we hope that the observed data are missing non-periodically. To that aim, we propose an intra-pulse frequency-coded FMCW (IPFC-FMCW) to solve the jamming problem.

The traditional FMCW signal causes periodic observation loss due to smart noise jamming. We propose an IPFC-FMCW signal, which consists of a train of FMCW chips with different widths and frequencies, with a random property to solve the problem of periodic data missing caused by jamming. In order to improve the performance of the waveform, the range profile of the waveform after jamming suppression is derived to evaluate the waveform performance. The optimization problem of waveform parameters is proposed, and the transmit waveform parameters are obtained by the simulated annealing method [27]. However, due to jamming, the problem of missing observations is inevitable. In the case of missing data, the problem of grid mismatch is evident in the grid-based CS method. We propose the refined-orthogonal matching pursuit (R-OMP) algorithm to obtain gridless range recovery results of scatterers.

In this paper, the proposed waveform is different from the conventional frequency coding waveform. On the one hand, some existing frequency coding waveforms are mainly used for detection under jamming-free conditions, where the constraints aim to obtain better autocorrelation performance, cross-correlation performance, etc. [28,29]. On the other hand, there are also some frequency coding waveforms used for anti-jamming. The frequency coding waveform in [22] varies the frequency coding of the transmit signal

against ISRJ, but it still suffers from periodically missing data due to the same width of the chip.

To sum up, the main contributions of this article are listed as follows.

(1) The periodic data missing of the FMCW signal in smart noise jamming scenarios causes a serious sidelobe problem. We propose the IPFC-FMCW waveform with non-periodic data missing in smart noise jamming scenarios to solve the periodic data missing problem of the FMCW signal.

(2) We take the three metrics of the range profile as the objective and use the simulated annealing method to optimize the waveform, which reduces the proportion of jamming signals and increases the peak to sidelobe ratio.

(3) Compared with the existing grid-based range recovery methods, the proposed range recovery method including a REFINE step is suitable for scenes with missing observations.

The rest of this paper is organized as follows. In Section 2, the working scenario and the transmitted signal model are introduced. In Section 3, the received signal model is introduced. In Section 4, the method of waveform design is introduced. In Section 5, the proposed range recovery approach is formulated. In Section 6, the simulation results evaluate the performance of waveform design and range recovery approach. Concluding remarks are presented in Section 7.

Throughout the article, we use $\mathbb{C}, \mathbb{R}$ to denote the sets of complex and real numbers, respectively. We denote the transpose, conjugate, and Hermitian transpose by $(\cdot)^T$, $(\cdot)^*$, and $(\cdot)^H$, respectively. The notations $\mathbb{C}^G$ and $\mathbb{C}^{N \times G}$ are the sets of $G$-dimensional vectors and $N \times G$ matrices of complex numbers, respectively.

## 2. Working Scenario and Transmitted Signal Model

In this section, we introduce the radar working scenario and transmitted signal model.

### 2.1. Working Scenario

In this subsection, we introduce the working scenario for a high-resolution radar. We consider a high-resolution radar observing closely placed scatterers. The working scenario is shown in Figure 1.

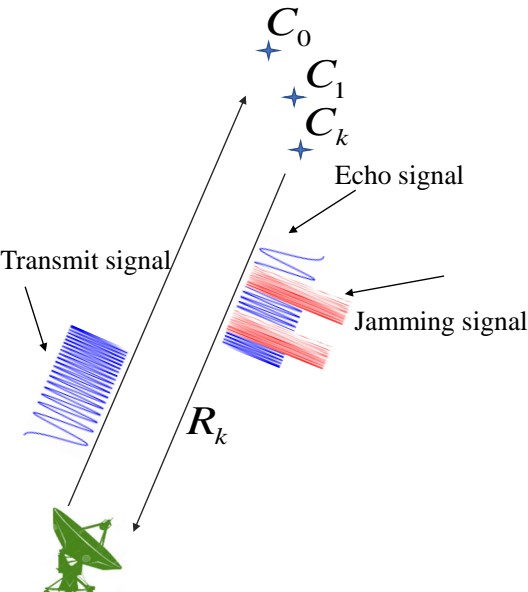

**Figure 1.** The working scenario of a high-resolution radar.

For single pulse scenes, the effect of Doppler can be neglected because the Doppler shift generated by radar movement or target movement is much smaller than the Doppler

tolerance [30–32]. Therefore, the Doppler effect is approximately negligible. For simplicity, in this paper, we assume that both scatterers and radar are stationary. The signal model and range recovery method are easy to be extended to the case of moving targets. Assume that the observation area consists of $K$ scatterers. The range of the $k$-th scatterer is denoted by $R_k$, for $k = 0, 1, \ldots, K - 1$. The scatterers' echo signal may be corrupted by the jamming signal from jammers, and the echo signal after jamming suppression has periodic or non-periodic observation missing. On the one hand, the signal is missing, and on the other hand, the sidelobe level increases, making it challenging to recover scatterers with high quality.

To obtain the high-resolution range recovery of these scatterers in the jamming scene, we employ the wideband IPFC-FMCW signal as the transmit signal. This kind of wideband waveform can achieve a high range resolution. By the width variation and frequency coding of the chip, this kind of waveform is also beneficial to solve the jamming problem. The transmitted signal model will be given in the following subsection.

### 2.2. Transmitted Signal Model

In this subsection, we formulate the expression of the transmitted waveform. We obtain a high-resolution range recovery in smart noise jamming using an IPFC-FMCW signal, which contains multiple FMCW chips with different widths and frequencies. From a practical point of view, these FMCW chips have an identical frequency modulation rate in order to reduce hardware complexity [33]. The proposed waveform is illustrated in Figure 2.

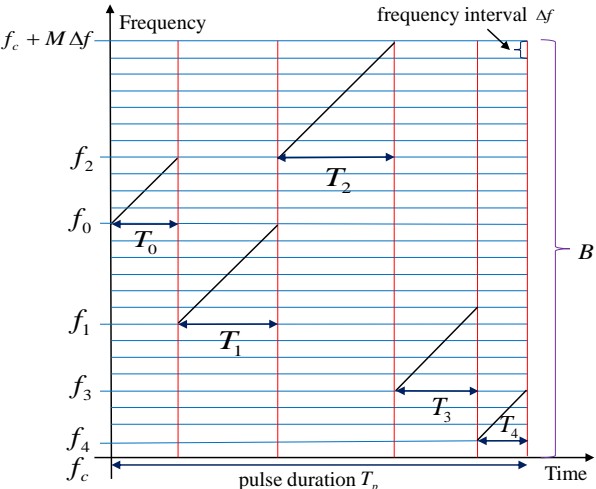

**Figure 2.** The proposed waveform consists of multiple FMCW chips with different widths and frequencies.

The transmit frequency set $\mathcal{F}$ consists of $M$ frequency codings, i.e., $\mathcal{F} = \{f_c + m\Delta f \mid m = 0, 1, \cdots, M - 1\}$, where $f_c$ is the minimal carrier frequency and $\Delta f$ is the frequency interval. Let $s_q(t)$ denote the baseband signal for the $q$-th chip, which is expressed as

$$s_q(t) = \text{rect}\left(\frac{t - T_q^s}{T_q}\right) e^{j\pi\alpha\left(t - T_q^s\right)^2}. \tag{1}$$

Here, $\alpha$ is the frequency modulation rate, $T_q$ represents the $q$-th chip width, and the window function is given by

$$\text{rect}(t) = \begin{cases} 1, & 0 \leq t \leq 1, \\ 0, & \text{otherwise}. \end{cases} \tag{2}$$

$T_q^s = \sum_{i=0}^{q-1} T_i$, which means that the signal transmission of the $q$-th chip begins at $t = \sum_{i=0}^{q-1} T_i$ and ends at $t = \sum_{i=0}^{q} T_i$. Let $Q$ be the number of chips, where $Q < M$. Then, the pulse duration $T_p$ is given by $T_p = \sum_{q=0}^{Q-1} T_q$.

Before the transmission of each chip, the FMCW baseband waveform is first modulated by the frequency chosen from $\mathcal{F}$. The transmitted waveform is given by

$$x(t) = \sum_{q=0}^{Q-1} \text{rect}\left(\frac{t - T_q^s}{T_q}\right) e^{j\pi\alpha\left(t - T_q^s\right)^2} e^{j2\pi f_q\left(t - T_q^s\right)}, \tag{3}$$

where $f_q$ is the frequency of the $q$-th chip. The maximum frequency of the set $\{f_q + \alpha T_q, q = 0, 1, \cdots, Q - 1\}$ is not greater than $f_c + M\Delta f$, so the maximum bandwidth of the transmitted waveform is expressed as $B = M\Delta f$.

## 3. Received Signal Model

The received signal of the radar is expressed as

$$\hat{y}(t) = \check{y}(t) + \check{i}(t) + \check{w}(t), \tag{4}$$

where $\check{y}(t)$, $\check{i}(t)$, and $\check{w}(t)$ denote the echo signal, jamming signal, and additive white Gaussian noise (AWGN), respectively. The echo signal model and jamming signal model are introduced below, respectively.

### 3.1. Echo Signal Model

In this subsection, we derive the expression of the echo signal. The round-trip delay between the transmitting antenna and the receiving antenna is $\tau_k = \frac{2R_k}{c}$ for the $k$-th scatterer, where $c$ is the speed of light. Assuming that the complex amplitude is the same within a pulse, the echo signal is expressed as

$$\tilde{y}(t) = \sum_{k=0}^{K-1} \beta_k x(t - \tau_k), \tag{5}$$

where $\beta_k$ is the complex amplitude of the $k$-th scatterer.

At the radar receiver, the echo signal $\tilde{y}(t)$ is mixed with the reference signal $x_{\text{ref}}(t)$ within each chip via the traditional dechirp processing [34]. Assuming that the reference distance is $R_{\text{ref}}$, the reference signal is expressed as $x_{\text{ref}}(t) = x(t - \tau_{\text{ref}})$, where $\tau_{\text{ref}} = 2R_{\text{ref}}/c$. Then, the output of each mixer is fed into a low-pass filter (LPF). For the $q$-th chip, the window length of the echo signal after dechirp processing is $T_q - |\tau_{\text{ref}} - \tau_k|$, as shown in Figure 3. Figure 3 shows the dechirp process.

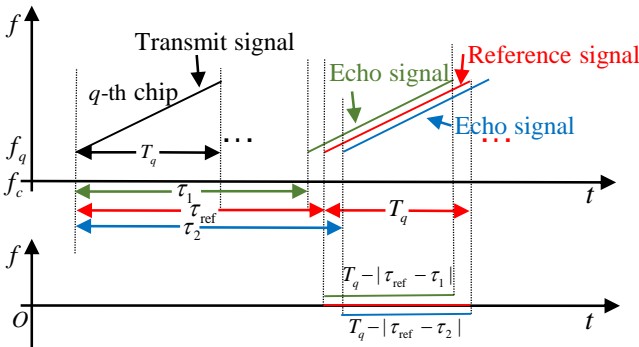

**Figure 3.** The dechirp process of the $q$-th chip.

Assuming that the range between the radar and the center of the target is $R_t$ for a target, its scatterer is close to the center of the target. So, we ignore the influence of different

scatterers on the window function. Then, the echo signal of all chips after the LPF is expressed as

$$\check{y}(t) \approx \sum_{q=0}^{Q-1} \text{rect}\left(\frac{t - T_q^s - \max\{\tau_{\text{ref}}, \tau_t\}}{T_q - |\tau_{\text{ref}} - \tau_t|}\right) \times \left(\sum_{k=0}^{K-1} \beta_k e^{j2\pi\alpha\left(t - T_q^s - \tau_{\text{ref}}\right)\bar{\tau}_k} e^{j2\pi f_q \bar{\tau}_k}\right), \quad (6)$$

where $\bar{\tau}_k = \tau_{\text{ref}} - \tau_k$, $\tau_t = 2R_t/c$. Then, the unknown delay is expressed as $\bar{\tau}_k = \frac{2\bar{R}_k}{c}$, where $\bar{R}_k = R_{\text{ref}} - R_k$.

### 3.2. Jamming Signal Model

In this subsection, we first introduce the jamming strategy of the jammer. Then, we provide the expression of the received jamming signal.

#### 3.2.1. Jamming Signal Generation

To interfere with the radar receiver, the jammer first intercepts the radar signal, then generates the jamming signal by modulating the intercepted radar signal, and finally forwards it to the radar receiver. Note that to improve the jamming efficiency, the jamming signal is not exactly the same as the intercepted signal but is generated by modulating the intercepted signal with a frequency-shifted signal. The jammer also modulates the frequency-shifted signal by multiplying with noise so as to form the blanket noise jamming effect in a specific range [16].

We next derive the model of smart noise jamming. For a self-defense jammer, the jammer is located on or close to the target, and the range between the radar and the jammer is $R_J$, where $R_J \approx R_t$. The $n_j$-th intercepted signal by the jammer is expressed as

$$i_{n_j}(t) = \text{rect}\left(\frac{t - n_j M_J T_J - \frac{R_J}{c}}{T_J}\right) x\left(t - \frac{R_J}{c}\right), \quad (7)$$

where $n_j = 0, 1, \cdots, N_J - 1$, $N_J = \lfloor \frac{T_p}{M_J T_J} \rfloor$ represents the number of slices intercepted by the jammer. For each intercepted slice, the jammer shifts the frequency of the intercepted signal according to the forwarding number $M_J - 1$ and multiplies it with noise [16]. For the $m_j$-th forwarding, since the lag time of the forwarding signal compared with the intercepted signal is $m_j T_J$, the frequency shift of the intercepted signal is expressed as $m_j \frac{T_J}{T_c} \Delta f$ so that a fake target with range $R_J$ can be formed. Then, the jammer forms smart noise jamming by multiplying the frequency-shifted signal with noise $n_a(t)$, which is a band-limited Gaussian signal with bandwidth $B_n$ and time width $T_J$. That is, the jammer generates smart noise jamming near the region with center $R_J$.

For the $m_j$-th forwarding of the $n_j$-th intercepted signal, the signal modulated by the jammer is represented as

$$i_{n_j,m_j}(t) = \text{rect}\left(\frac{t - n_j M_J T_J - m_j T_J - \frac{R_J}{c}}{T_J}\right) x\left(t - m_j T_J - \frac{R_J}{c}\right) \times \beta_J e^{j2\pi m_j \frac{T_J}{T_c} \Delta f\left(t - n_j M_J T_J - m_j T_J - \frac{R_J}{c}\right)} \times n_a(t), \quad (8)$$

where $m_j = 1, 2, \cdots, M_J - 1$, $\beta_J$ represents the amplitude of the jamming signal. The working process of the jammer is shown in Figure 4.

Figure 4a represents the interception process of the jammer, which indicates the time-frequency diagrams of the radar transmit signal and the intercepted signal by the jammer, respectively. Figure 4b represents the modulation and forwarding process of the jammer, which modulates the intercepted signal and multiplies it with noise to obtain smart noise jamming, and the time-frequency diagram of the jamming signal is shown in the purple part.

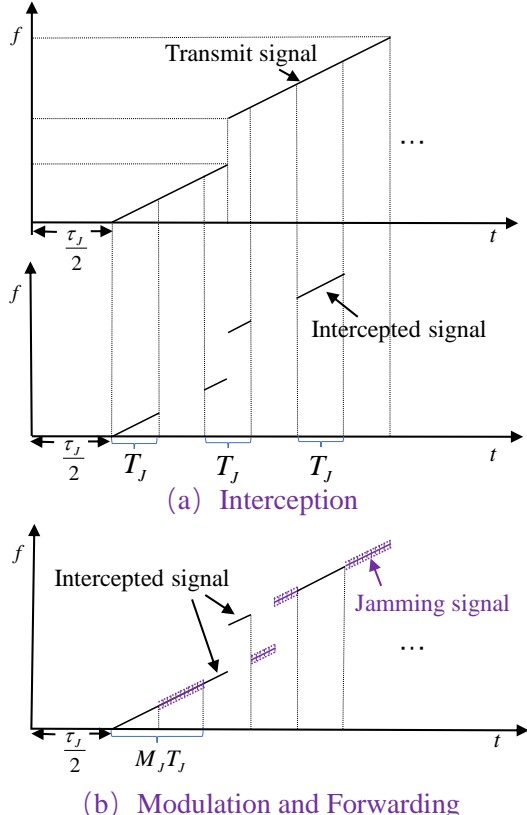

Figure 4. The working process of the jammer.

### 3.2.2. Received Jamming Signal

The jamming signal received by the radar is expressed as

$$
\begin{aligned}
\tilde{i}(t) &= \sum_{n_j=0}^{N_J-1} \sum_{m_j=1}^{M_J-1} i_{n_j,m_j}\left(t - \frac{R_J}{c}\right) \\
&= \sum_{n_j=0}^{N_J-1} \sum_{m_j=1}^{M_J-1} \operatorname{rect}\left(\frac{t - n_j M_J T_J - m_j T_J - \tau_J}{T_J}\right) x\left(t - m_j T_J - \tau_J\right) \times \beta_J e^{j2\pi m_j \frac{T_J}{T_c}\Delta f\left(t - n_j M_J T_J - m_j T_J - \tau_J\right)} \times n_a(t),
\end{aligned} \tag{9}
$$

where $\tau_J = \frac{2R_J}{c}$.

Similarly, the jamming signal is mixed with the reference signal and fed into the LPF. The jamming signal is $\breve{i}(t) = \operatorname{LPF}\left(\tilde{i}(t) \cdot x_{\text{ref}}^*(t)\right)$. Figure 5 shows the dechirp and LPF process of the jamming signal.

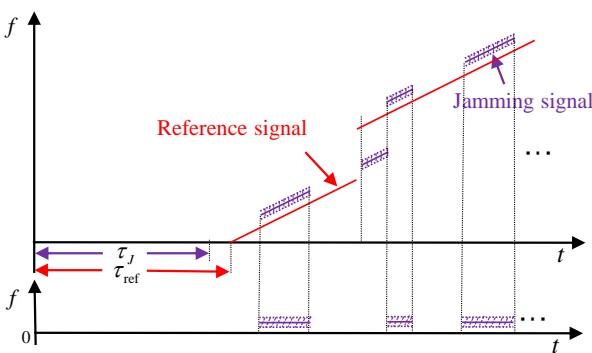

Figure 5. The jamming signal processing process.

In conclusion, the proposed waveform and the conventional FMCW behave differently in jamming scenarios. We compare the periodic jamming for the FMCW signal with the non-periodic jamming for the proposed waveform by a schematic diagram as shown in Figure 6. The proposed IPFC-FMCW signal with non-periodic jamming is better than the FMCW signal with periodic jamming in smart noise jamming.

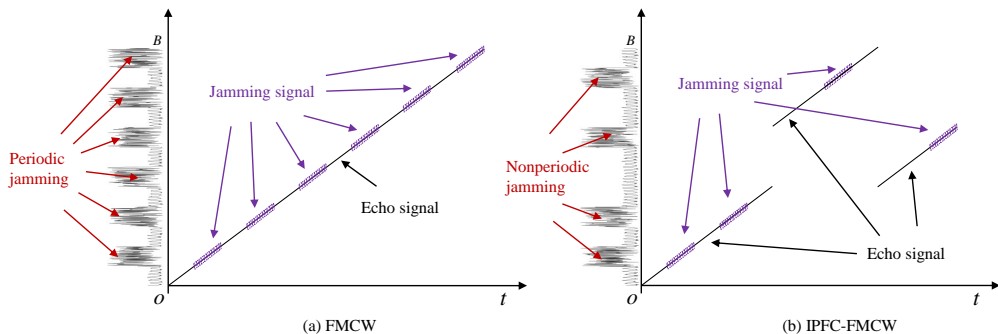

**Figure 6.** Comparison of two waveforms with smart noise jamming.

## 4. Waveform Design

For the proposed waveform, how to obtain the waveform parameters is the problem to be solved. In this section, we first present the signal model after jamming suppression. Then, we formulate the optimization problem with respect to the waveform parameters. Finally, we propose a method based on simulated annealing to obtain the waveform parameters.

### 4.1. Jamming Suppression

In general, the power of the jamming signal is much higher than that of the echo signal, and the jamming signal overlaps with the echo signal in the frequency domain. We perform signal elimination on the part of the jamming signal and the echo signal that overlap in the frequency domain. The jamming suppression diagram is shown in Figure 7.

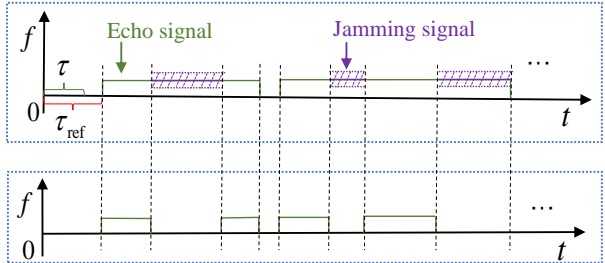

**Figure 7.** The principle diagram of the jamming suppression.

According to (6), the window function of the echo signal is expressed as

$$h^e(t, \tau) = \sum_{q=0}^{Q-1} \text{rect}\left(\frac{t - T_q^s - \max\{\tau_{\text{ref}}, \tau_{\text{ref}} - \tau\}}{T_q - |\tau|}\right), \tag{10}$$

where $\tau = \tau_{\text{ref}} - \tau_t$. For the part where the jamming signal overlaps with the echo signal in the frequency domain, the echo signal is submerged in the jamming signal because the amplitude of the jamming signal is much higher than that of the echo signal. Usually, a threshold is used to detect the jamming part. After the jamming detection, the jamming

suppression is equivalent to adding windows to the received signal, where the window function is expressed as

$$h(t, \tau) := \begin{cases} 0, & |\bar{y}(t)| > \epsilon \cdot \text{mean}\{|\bar{y}(t)|\}, \\ h^e(t, \tau), & \text{otherwise}. \end{cases} \tag{11}$$

where the label mean$\{\cdot\}$ indicates the mean value of $\{\cdot\}$ within a pulse, and $\epsilon$ is a parameter. For a target, the signal after dechirp and jamming suppression is expressed as

$$x^{\text{supp}}(t, \tau) = h(t, \tau) \sum_{q=0}^{Q-1} \left( \frac{t - T_q^s - \max\{\tau_{\text{ref}}, \tau_{\text{ref}} - \tau\}}{T_q - |\tau|} \right) \times e^{j2\pi\alpha(t - T_q^s - \tau_{\text{ref}})\tau} e^{j2\pi f_q \tau}. \tag{12}$$

*4.2. Waveform Design Problem Formulation*

In the previous subsection, the signal expression after jamming suppression is given, which means that the observed data are partially missing. Since data missing will reduce the signal, widen the main lobe, and enlarge the side lobe, we use the range profile to measure the waveform performance after jamming suppression and expect to optimize waveform parameters to improve the radar performance.

In the tracking scenario, the received signal after jamming suppression is denoted as $x^{\text{supp}}(t, 0)$ for an ideal scatter target. The range profile of the received signal after jamming suppression is expressed as

$$\chi(\tau) = \int_{-\infty}^{+\infty} x^{\text{supp}}(t, 0) x^{\text{supp}*}(t, \tau) dt, \tag{13}$$

which illustrates the performance of correlation with different delays. Here, the range profile reflects the signal-to-noise ratio loss (SNRL), main lobe width (MLW) and peak to sidelobe ratio (PSLR). SNRL refers to the loss of signal-to-noise ratio in the range profile compared to the jamming-free signal. Jamming suppression reduces the observed signal and increases the SNRL of the range profile, and we hope for the SNRL of the range profile to be as small as possible. In order to maintain the high range resolution of the wideband signal, we also hope to limit the MLW of the range profile to a certain range, namely $\tau_{\text{res}} \leq \text{MLW}(\chi(\tau)) \leq \gamma_{\text{mlw}} \tau_{\text{res}}$, where $\tau_{\text{res}} = \frac{c}{2B}$ represents the ideal range resolution, $\gamma_{\text{mlw}} \geq 1$. The observed data are missing, which causes the increase of sidelobe level, and we want the PSLR to be as high as possible to reduce the sidelobe level. In order to reduce the proportion of jamming, constrain the sidelobe level and limit the main lobe width, we select the SNRL, MLW, and PSLR as the optimization objectives. Next, waveform parameters are designed to achieve a good range profile performance after jamming suppression. Let $t$ denote the vector variable consisting of the widths of FMCW chirps, namely

$$t = [T_0, T_1, \cdots, T_{Q-1}]. \tag{14}$$

Let $f^c$ denote the frequencies of the chips of the transmit signal, which is expressed as

$$f^c = [f_0, f_1, \cdots, f_{Q-1}]. \tag{15}$$

The IPFC-FMCW design problem is then formulated as

$$
\begin{aligned}
\min_{t, f^c} \quad & l = \text{SNRL}(\chi(\tau)) - \gamma \text{PSLR}(\chi(\tau)) - l_{\text{mlw}} \text{rect}\left( \frac{\text{MLW}(\chi(\tau)) - \tau_{\text{res}}}{(\gamma_{\text{mlw}} - 1)\tau_{\text{res}}} \right) \\
\text{s.t.} \quad & f_q \in \mathcal{F}, \ f_q + \alpha T_q \leq f_c + B, \ q = 0, 1, \cdots, Q - 1, \\
& \sum_{q=0}^{Q-1} T_q = T_p,
\end{aligned} \tag{16}
$$

where $\gamma$ represents the compromise coefficient between SNRL and PSLR of the objective function. Then, $\gamma_{\text{mlw}}$ represents the MLW coefficient after jamming suppression, which indicates the range resolution of the waveform after jamming suppression is in a certain range of $\tau_{\text{res}} \leq \text{MLW}(\chi(\tau)) \leq \gamma_{\text{mlw}}\tau_{\text{res}}$. The $l_{\text{mlw}}$ is a large constant that satisfies the constraint of MLW.

*4.3. Waveform Design Method*

Equation (16) wields optimization over waveform parameters to find the width and frequency coding for each chip of the transmitted signal. The optimizations are non-linear and require a multi-dimensional parameter search. The simulated annealing algorithm is a global random search method based on the evolution of survival of the fittest, natural selection, and population genetic evolution in nature [27]. It is widely used in automatic control, pattern recognition, engineering design, and intelligent fault diagnosis to solve complex non-linear and multi-dimensional optimization problems. So, we use the simulated annealing algorithm to optimize the waveform parameters.

For the problem of waveform parameters optimization, we first randomly choose a set of waveform parameters conforming to the constraint conditions in (16) as the initial values, and then, we use the simulated annealing algorithm to obtain the waveform parameters $t$, $f^c$ of the transmit waveform under a specific jamming strategy. The process of the simulated annealing algorithm is described in Algorithm 1. This demonstrates a heuristic random search process. As the temperature decreases, we obtain the waveform parameters.

---

**Algorithm 1** Simulated Annealing

---

**Input:** Initial temperature $T_{\text{max}}$, termination temperature $T_{\text{min}}$, annealing coefficient $\mu$, $K_{\text{marcov}}$, $T = T_{\text{max}}$. Randomly generate the width and frequency coding of each chip conforming to the constraints of (16), denoted as $t_{\text{currest}}$, $f^c_{\text{currest}}$. Initialize the evaluation value $l_{\text{currest}}$.

**Steps:**
1: **while** $T > T_{\text{min}}$ do
2:    **for** $i = 1, \cdots, K_{\text{marcov}}$
3:       Two chips are randomly selected from $t_{\text{currest}}$ and $f^c_{\text{currest}}$, and the width and frequency of these two chips are modified to obtain the new $t_{\text{new}}$, $f^c_{\text{new}}$ that satisfies the constraints of (16).
4:       Calculate the evaluation value $l_{\text{new}}$ corresponding to $t_{\text{new}}$, $f^c_{\text{new}}$.
5:       **if** $l_{\text{new}} < l_{\text{currest}}$
6:          $t_{\text{currest}} = t_{\text{new}}, f^c_{\text{currest}} = f^c_{\text{new}}, l_{\text{currest}} = l_{\text{new}}$
7:          **if** $l_{\text{new}} < l_{\text{best}}$
8:             $t_{\text{best}} = t_{\text{new}}, f^c_{\text{best}} = f^c_{\text{new}}, l_{\text{best}} = l_{\text{new}}$
9:          **end**
10:       **else** $l_{\text{new}} > l_{\text{currest}}$
11:          Generate random number $\eta \in [0, 1]$ to compare with $\exp(-(l_{\text{new}} - l_{\text{currest}})/T)$.
12:          **if** $\eta < \exp(-(l_{\text{new}} - l_{\text{currest}})/T)$
13:             $t_{\text{currest}} = t_{\text{new}}, f^c_{\text{currest}} = f^c_{\text{new}}, l_{\text{currest}} = l_{\text{new}}$
14:          **else**
15:             $t_{\text{new}} = t_{\text{currest}}, f^c_{\text{new}} = f^c_{\text{currest}}$
16:          **end**
17:       **end**
18:    **end for**
19:    $T = T \times \mu$
20: **end while**

**Output:** $t_{\text{best}}, f^c_{\text{best}}, l_{\text{best}}$.

---

## 5. Range Recovery Approach

In this section, we first provide the signal model after jamming suppression. Then, we introduce a new signal processing method to recover parameters of scatterers from the

radar echoes after jamming suppression. Compared with the existing grid-based range recovery methods, the proposed range recovery method including a REFINE step is suitable for scenes with missing observations.

### 5.1. Problem Formulation

First, the received signal is uniformly sampled with rate $F_s = \frac{1}{T_s}$, where $T_s$ is the sampling interval. In particular, the sampling rate $F_s$ is set to $F_s = \frac{2R_{\max}B}{cT_p}$, where $R_{\max}$ is the maximum detection range. The number of sample points is $G = \lfloor \frac{T_p}{T_s} \rfloor$, and the sample time instances are $t = \tau_{\mathrm{ref}} + gT_s$, where $g \in \{0, 1, \cdots, G-1\}$. Then, the jamming suppression is performed on the received signal to obtain the signal model after jamming suppression, and the sampled signal after jamming suppression is given by

$$\bar{y}[g] = h[g] \left( \sum_{k=0}^{K-1} \beta_k e^{j2\pi \left( f_q + \alpha(gT_s - T_q^s) \right) \bar{\tau}_k} + \check{w}[g] \right), gT_s \in \left[ T_q^s, T_{q+1}^s \right], \tag{17}$$

where $\check{w}[g]$ is the discrete-time Gaussian noise, and $h[g] = h(\tau_{\mathrm{ref}} + gT_s, \tau) \in \{0, 1\}$ indicates whether the echo is located at the $g$-th sample. The frequency describing the unknown parameter $\bar{\tau}_k$ is expressed as $\left( f_q + \alpha(gT_s - T_q^s) \right)$ in (17).

Next, we present the non-zero observed data. For the unknown parameter $\bar{\tau}_k$, the frequency of the $g$-th sample is expressed as $\bar{f}[g] = \left( f_q + \alpha(gT_s - T_q^s) \right), gT_s \in \left[ T_q^s, T_{q+1}^s \right]$. Denote the frequency vector of sampled data by $\bar{f} = [\bar{f}[0], \bar{f}[1], \cdots, \bar{f}[G-1]]^T \in \mathbb{R}^G$. Since $h[g] \in \{0, 1\}$, there are many zeros in the observed data, and we construct an effective frequency vector. For the sample $h[g] = 0$, we first remove the frequency $\bar{f}[g]$ from $\bar{f}$; then, we arrange the remaining frequency in an ascending order to obtain the effective frequency vector $f = [f[0], f[1], \cdots, f[\bar{G}-1]] \in \mathbb{R}^{\bar{G}}$, where $\bar{G}$ represents the number of non-zero observations after jamming suppression. The sampled signal after jamming suppression is obtained by using the same method, which is expressed as

$$y[\bar{g}] = \sum_{k=0}^{K-1} \beta_k e^{j2\pi f[\bar{g}] \bar{\tau}_k} + w[\bar{g}], \tag{18}$$

where $w[\bar{g}]$ is constructed by $\check{w}[g]$ in the same way as $y[\bar{g}]$ from $\bar{y}[g]$. We denote by $y = [y[0], y[1], \cdots, y[\bar{G}-1]]^T \in \mathbb{C}^{\bar{G}}$ the received signal. The task of high-resolution radar is to recover the range of the scatterers, which can be recovered by estimating the values of $\{\bar{R}_k\}$ from $\bar{\tau}_k$.

### 5.2. Range Estimation Method

The subsection introduces the proposed R-OMP algorithm that estimates the ranges of multiple scatterers.

In this subsection, we estimate the delay $\bar{\tau}_k$ by using the received signal. We then infer the corresponding range $\bar{R}_k$ from the delay estimation.

The received signal is expressed as

$$y = \sum_{k=0}^{K-1} \beta_k a(\bar{\tau}_k) + w, n = 0, 1, \cdots, N-1, \tag{19}$$

where $\bar{\tau}_k$ and $\beta_k$ are the time delay parameter and complex scattering coefficient of the $k$-th scatterer, respectively, $a(\bar{\tau}_k) = [e^{j2\pi f[0] \bar{\tau}_k}, e^{j2\pi f[1] \bar{\tau}_k}, \cdots, e^{j2\pi f[\bar{G}-1] \bar{\tau}_k}]^T \in \mathbb{C}^{\bar{G}}$ is the steering vector, and $w$ is constructed by $w[\bar{g}]$ in the same way as $y$ from $y[\bar{g}]$.

In this part, we develop an iterative algorithm to estimate the range, denoted as R-OMP, which operates over the continuous-valued parameter space [9]. For each iteration,

we use the grid-based CS algorithm to obtain rough estimates of scatterers and then obtain the accurate estimation of scatterers through refinement [35].

To recover the radar scatterers, the delay is first discretized by intervals equal to the range resolution, which is expressed as $\tau_{\text{res}}$. Thus, the grid set of discretized is denoted by $\bar{\tau} := \{\bar{\tau}^0, \bar{\tau}^1, \cdots, \bar{\tau}^{U-1}\}$. Assuming that the scatterers are located on these grids, the scatterer scene can be indicated by $\boldsymbol{\beta} \in \mathbb{C}^U$ with entries

$$[\boldsymbol{\beta}]_u := \begin{cases} \beta_k, & \text{if there exists } \bar{\tau}_k = \bar{\tau}^u, \\ 0, & \text{otherwise.} \end{cases} \tag{20}$$

Following (19), it holds that

$$\boldsymbol{y} = \boldsymbol{A}\boldsymbol{\beta} + \boldsymbol{w}, \tag{21}$$

where $\boldsymbol{A} \in \mathbb{C}^{G \times U}$ is the steering matrix, the entries of which are given by $[\boldsymbol{A}]_{:,u} = \boldsymbol{a}(\bar{\tau}^u)$.

The first part calculates the rough estimation of the parameters. Assume that before the $k$-th iteration, we obtain the estimation results $\hat{\Omega}_{k-1} = \{\hat{\beta}_i, \hat{\tau}_i\}_{i=0}^{k-1}$ from the previous $k-1$ iterations. Firstly, the residual signal is calculated according to the received signal and the estimated results of $k-1$ scatterers, which is expressed as

$$\boldsymbol{P}_{\hat{\Omega}_{k-1}}^{\perp}\boldsymbol{y} = \boldsymbol{y} - \boldsymbol{A}^{k-1}\left((\boldsymbol{A}^{k-1})^H\boldsymbol{A}^{k-1}\right)^{-1}(\boldsymbol{A}^{k-1})^H\boldsymbol{y}, \tag{22}$$

where $\boldsymbol{A}^{k-1} = [\boldsymbol{a}(\hat{\tau}_0), \boldsymbol{a}(\hat{\tau}_1), \cdots, \boldsymbol{a}(\hat{\tau}_{k-1})]$ represents the steering matrix composed of $k-1$ scatterers. Then, the rough estimate of the $k$-th scatterer is expressed as

$$\{\hat{\tau}_k\} = \underset{\bar{\tau}}{\operatorname{argmax}} \left| \boldsymbol{A}^H \boldsymbol{P}_{\hat{\Omega}_{k-1}}^{\perp} \boldsymbol{y} \right|, \tag{23}$$

$$\hat{\beta}_k = \left( \boldsymbol{a}^H(\hat{\tau}_k)\boldsymbol{a}(\hat{\tau}_k) \right)^{-1} \boldsymbol{a}^H(\hat{\tau}_k)\boldsymbol{y}. \tag{24}$$

Then, The next two positions should add an indentation. the parameter set of the first $k$ scatterers is expressed as

$$\hat{\Omega}_k = \hat{\Omega}_{k-1} \cup \{\hat{\beta}_k, \hat{\tau}_k\}. \tag{25}$$

In fact, the time delay $\bar{\tau}_k$ of the scatterer is not always at the grid set, and there is a grid mismatch. Next, we refine the parameter set to obtain accurate estimation results.

The second part refines the parameters by gradient descent using backtracking line search. The REFINE step aims to find the maximum likelihood estimate by minimizing the loss function using iterative gradient descent through the rough estimation $\hat{\Omega}_k$ obtained at the first part as initialization. The loss function is defined according to the rough estimation parameters $\hat{\Omega}_k$, expressed as

$$l(\hat{\Omega}_k) = \frac{1}{2}\left\| \boldsymbol{y} - \sum_{i=0}^{k} \hat{\beta}_i \boldsymbol{a}(\hat{\tau}_i) \right\|^2. \tag{26}$$

Then, we obtain the $\{\hat{\beta}_k, \hat{\tau}_k\}_{k=0}^{K-1}$. We summarize the proposed algorithm in Algorithms 2 and 3. The range estimate $\hat{R}_k$ is expressed as $\hat{R}_k = \frac{c\hat{\tau}_k}{2}$.

---

**Algorithm 2** R-OMP

---

**Input:** signal $y$, sparsity $K$, and parameter $\hat{\Omega}_{-1} = \varnothing$
**for** $k = 0 : K - 1$
  1: Compute the $P_{\hat{\Omega}_{k-1}}^{\perp} y$
  2: Find the parameter $\{\hat{\tau}_k\}$ of the $k$-th scatterer
  3: Find the parameter $\{\hat{\beta}_k\}$ of the $k$th scatterer
  4: Update : $\hat{\Omega}_k = \hat{\Omega}_{k-1} \cup \{\hat{\beta}_k, \hat{\tau}_k\}$
  5: $\hat{\Omega}_k \leftarrow \text{REFINE}(y, \hat{\Omega}_k)$.
**Output:** $\hat{\Omega}_K$.

---

**Algorithm 3** REFINE$(y, \Omega)$

---

**Input:** Parameter $\alpha_0, \eta \in (0, 0.5), \mu \in (0, 1)$
**repeat:**
  1: **for** each $\Omega^i$ in $\Omega$
  2:     Compute loss function $l(\Omega)$ and gradient $\nabla l(\Omega)$
  3:     $\alpha \leftarrow \alpha_0, \tilde{\Omega} \leftarrow \Omega$
  4:     **repeat:**
  5:        Update $\Omega^i \leftarrow \Omega^i - \alpha \nabla l(\Omega^i), \alpha \leftarrow \alpha \times \mu$
  6:     **until:** $l(\Omega) \leq l(\tilde{\Omega}) - \eta \alpha (\nabla l(\Omega^i))^2$
  7: **end for**
**until convergence**

---

## 6. Simulations

In this section, we set the simulation parameters and present the evaluation results of waveform design and range recovery approach.

### 6.1. Simulation Settings

For convenience, we summarize the main parameters used in the experimental evaluation in Table 1.

**Table 1.** The simulation parameters.

| Parameter | Value |
|---|---|
| Radar initial frequency $f_c$ | 8 GHz |
| Radar pulse width $T_p$ | 50 μs |
| Radar signal bandwidth $B$ | 2 GHz |
| Number of carrier frequency $M$ | 50 |
| Frequency interval $\Delta f$ | 40 MHz |
| Number of chips $Q$ | 8 |
| Tracking delay error $\tau_{\text{ref}} - \tau_t$ | 0 μs |
| Radar sampling frequency $f_s$ | 40 MHz |
| Jammer sampling period $T_J$ | 1 μs |
| Jammer duty cycle parameter $M_J$ | 2 |
| Number of scatterers $K$ | 4 |
| Scatterers amplitude $\beta_k$ | {1, 1, 0.32, 0.18} |
| Scatterers ranges $\bar{R}_k$ | {3, 4, 5, 6.87} m |
| Monte Carlo trials $M_c$ | 500 |

In the next discussion, we use FMCW (ideal) to denote an ideal FMCW signal without jamming. We give the range profile results of different waveforms to illustrate the validity of the proposed waveform. We use the hit rate as the performance criterion to demonstrate the radar performance in different noise levels for different waveforms and methods. A "hit" is defined if a scatterer's range is successfully recovered. Each hit rate is calculated

over $M_c$ Monte Carlo trials by recovering the scatterers depicted in Table 1. The root mean square error (RMSE) of range estimation is used to evaluate the estimation accuracy and compared with Cramer–Rao bound (CRB) [36]. Particularly, we carry out $M_c$ Monte Carlo trials and denote the RMSE of $\{\bar{R}\}$ by

$$\text{RMSE of } \bar{R}_k : \sqrt{\frac{1}{M_c} \sum_{i=0}^{M_c-1} \left( \bar{R}_k - \hat{R}_k^i \right)^2}, \tag{27}$$

where $\hat{R}_k^i$ represents the range estimation results of the *k*-th scatter for the *i*-th Monte Carlo trial. In addition, the range recovery results of different waveforms and different algorithms are also given, which intuitively show the recovery performance of different waveforms and different algorithms for different scatterers.

### *6.2. Waveform Design Evaluation*

In this subsection, we carry out simulations to verify the effectiveness of the proposed waveform design method. First we introduce the choice of coefficients for the waveform design optimization problem, and then the waveform design evaluation is conducted in two kinds of jamming scenes. In the first scene, the jamming parameters are estimated, while in the second scene, the jamming parameters are unknown. The proposed IPFC-FMCW waveform with random parameters is compared with the ideal FMCW waveform without a jamming signal to show the effectiveness of waveform parameters design.

#### 6.2.1. Optimization Problem Coefficient

For the jamming scene in Table 1, the SNRL and PSLR are obtained in different $\gamma_{\text{mlw}}$ against various $\gamma$, as shown in Figure 8. From the figure, it is seen that the SNRL and PSLR are fluctuating in different $\gamma$, which is because the optimization space is discrete. As the $\gamma_{\text{mlw}}$ factor becomes larger, PSLR becomes larger. When the coefficients of the optimization problem are within a certain range, we obtain good waveform performance. In the next simulation, the compromise coefficient $\gamma = 1$ between SNRL and PSLR of the objective function, the constraint coefficient $l_{\text{mlw}}$ and the MLW coefficient $\gamma_{\text{mlw}}$ are set as 100 and 4/3, respectively.

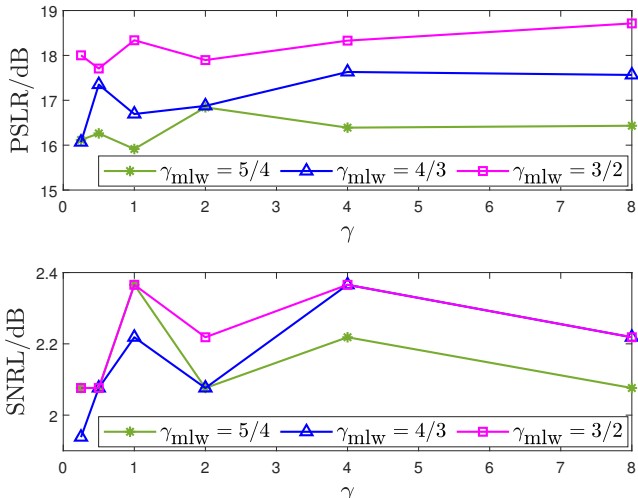

**Figure 8.** SNRL, PSLR versus different $\gamma$.

#### 6.2.2. Jamming with Estimated Parameters

We consider the scenario where the jamming parameters are estimated. The jamming parameters are shown in Table 1. The waveform parameter design under a specific jamming strategy is considered.

First, we compare the SNRL, PSLR and MLW of the optimized waveform with other waveforms, and the results are shown in Table 2. It can be seen from the table that the PSLR of FMCW waveform is relatively poor because the FMCW waveform is periodically missing in the frequency domain due to jamming, and the grating problem is severe. The IPFC-FMCW waveform with optimized parameters has better performance than the IPFC-FMCW waveform with random parameters.

**Table 2.** Performance comparison with different waveform when jamming parameters are known.

|              | SNRL (dB) | PSLR (dB) | MLW (m) |
|--------------|-----------|-----------|---------|
| FMCW (ideal) | 0         | 13.25     | 0.075   |
| FMCW         | 3.01      | 3.92      | 0.075   |
| Random       | 2.68      | 6.96      | 0.1     |
| Optimized    | 2.22      | 16.69     | 0.096   |

Then, we compare the range profile of the optimized waveform with other waveforms as shown in Figure 9. We use the ideal FMCW waveform without jamming as a reference. As we can see, the amplitude of the peak of the black line is 0 dB, indicating the peak of an ideal FMCW signal with no signal loss. The peaks of several other waveforms are below 0 dB, indicating the signals are missing due to jamming. The proposed waveform with optimized parameters has the least signal loss. For sidelobe levels, the proposed waveform with optimized parameters has a smaller sidelobe level than other waveforms. The IPFC-FMCW waveform with optimized parameters has a smaller SNRL and a lower sidelobe compared with the FMCW waveform and the IPFC-FMCW waveform with random parameters.

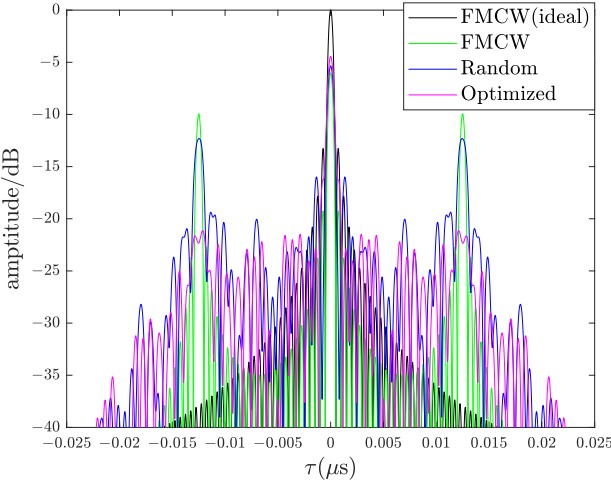

**Figure 9.** Range profile comparison of different waveforms when the jamming parameters are estimated.

### 6.2.3. Jamming with Unknown Parameters

We consider the scenario with unknown jamming parameters. Because of previous observations, the jamming parameters are obtained from historical jamming parameters set $\mathcal{T}_{\mathcal{J}}$ and $\mathcal{M}_{\mathcal{J}}$, i.e., $\mathcal{T}_{\mathcal{J}} = \{1\ \mu s, 2\ \mu s\}$, and $\mathcal{M}_{\mathcal{J}} = \{2, 3\}$. The design of waveform parameters is considered under the uniform distribution of different typical jamming parameters.

First, we compare the SNRL, PSLR and MLW of the optimized waveform with other waveforms, and the results are shown in Table 3. It can be seen that the PSLR of FMCW waveform is relatively poor, and the IPFC-FMCW waveform with optimized parameters has better performance than the IPFC-FMCW waveform with random parameters.

**Table 3.** Performance comparison with different waveform when jamming parameters are unknown.

|              | SNRL (dB) | PSLR (dB) | MLW (m) |
|--------------|-----------|-----------|---------|
| FMCW (ideal) | 0         | 13.25     | 0.075   |
| FMCW         | 3.67      | 2.86      | 0.075   |
| Random       | 2.4       | 8.9       | 0.097   |
| Optimized    | 2.01      | 12.83     | 0.082   |

Then, we compare the range profile of the optimized waveform with other waveforms under different jamming parameters. For each jamming scenario, the IPFC-FMCW waveform with optimized parameters has a smaller SNRL and a lower sidelobe compared with FMCW waveform and the IPFC-FMCW waveform with random parameters, as shown in Figure 10a–d.

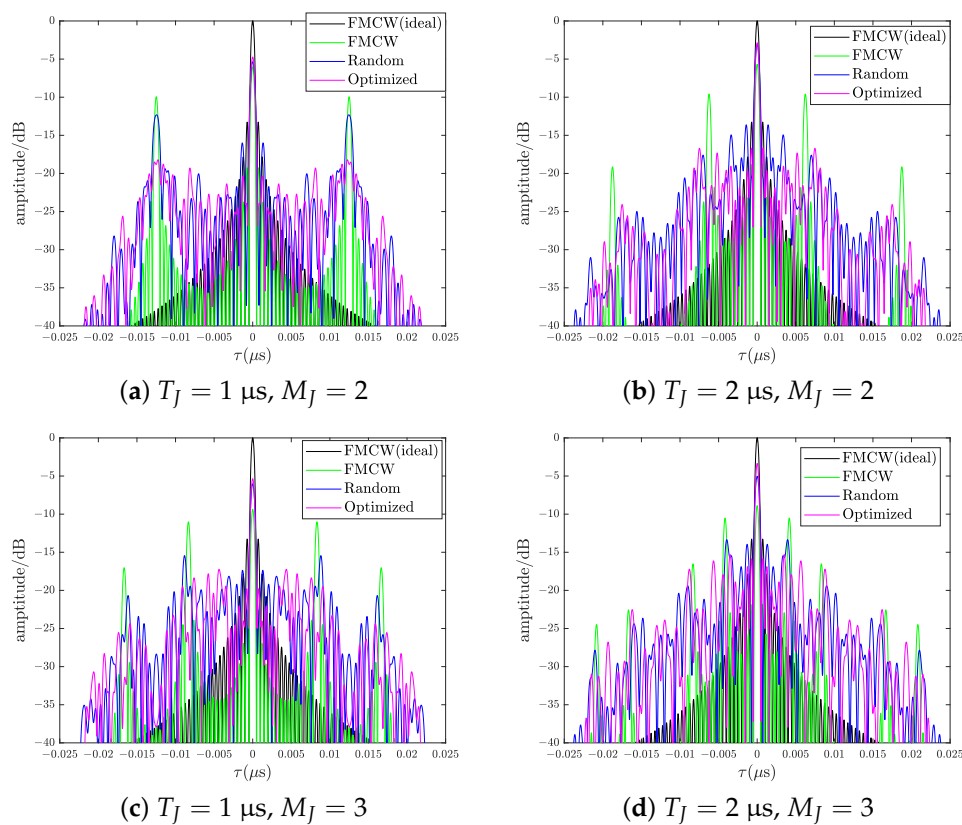

**(a)** $T_J = 1$ μs, $M_J = 2$

**(b)** $T_J = 2$ μs, $M_J = 2$

**(c)** $T_J = 1$ μs, $M_J = 3$

**(d)** $T_J = 2$ μs, $M_J = 3$

**Figure 10.** Range profile comparison of different waveforms under different jamming parameters.

*6.3. Range Recovery Approach Evaluation*

In this subsection, we perform simulations to verify the effectiveness of the proposed range estimation method. The ranges and amplitudes of scatterers are shown in Table 1, respectively. The powers of the first and second scatterers are the same, and the powers of the third and fourth scatterers are 10 dB and 15 dB lower than the power of the first scatterer, respectively.

First, the range recovery results of a single-trial are shown in Figure 11a–d, where the signal-to-noise ratio (SNR) of the first scatterer is set to 10 dB. The SNR is defined as $SNR = 10\log_{10}\frac{|\beta|^2}{\sigma^2}$, where $\beta$ and $\sigma^2$ indicate the amplitude of the echo signal and the power of the AWGN, respectively. Matched filter (MF) technology calculates the correlation of the received echo and the transmitted signal with different delays [37]. Compared with the MF algorithm, the estimation result of the proposed algorithm is closer to the true point. The comparison of the existing MF and our method shows the superiority of the proposed

method in the performance of range recovery. In addition, the reconstruction results of the optimized waveform are better than the results of other waveforms.

Then, we simulate the performance of scatterer recovery versus different radar SNR. The hit rate performance is utilized as the performance criterion. Each hit rate is calculated over 500 Monte Carlo trails by recovering the locations of the four scatterers depicted in Table 1. The hit rate versus SNR of the first scatterer is shown in Figure 12. Observing Figure 12, we note that the hit rate of the R-OMP method of the IPFC-FMCW waveform with random parameters, IPFC-FMCW waveform with optimized parameters and FMCW waveform without jamming reach a probability of one with the increasing of SNR, and the hit rate based on the R-OMP method for FMCW waveform does not reach a probability of one because the fourth scatterer is not detected due to the high sidelobe level. Figure 12 also shows that the waveform with optimized parameters has better performance than the waveform with random parameters. In addition, compared with the MF method [37], the R-OMP method has a higher hit rate and better performance.

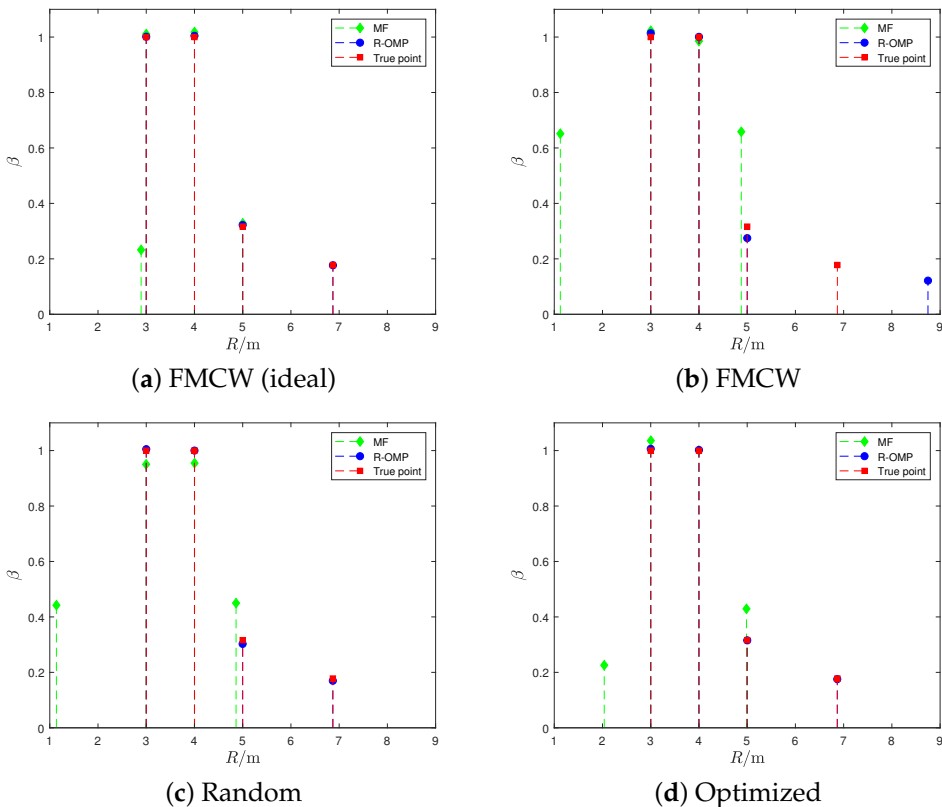

**Figure 11.** Recovery result of range parameter.

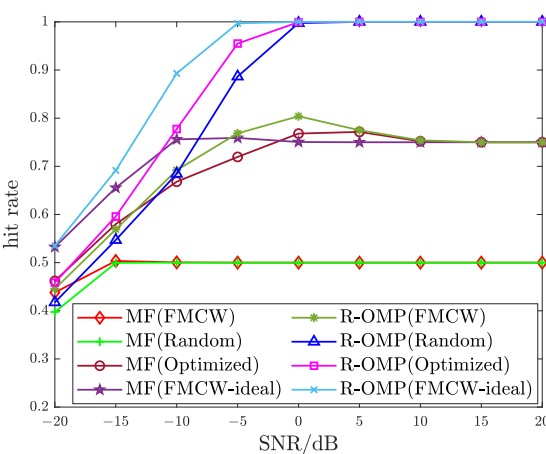

**Figure 12.** Range recovery hit rate versus different SNR.

Next, we compare the RMSE of range estimation with other methods in the strong and weak scatterer for the different waveforms. The RMSE are obtained through 500 Monte Carlo trials. The RMSE values of the range estimate obtained via different algorithms and waveforms are compared with the corresponding CRB against various SNR, as shown in Figures 13 and 14. From Figure 13, we see that the proposed method asymptotically reaches CRB, illustrating the validity of the proposed method. It can be seen from Figure 14 that the traditional FMCW waveform has a high sidelobe due to jamming, and the proposed waveform has a better PSLR, so that the weak scatterer is estimated effectively.

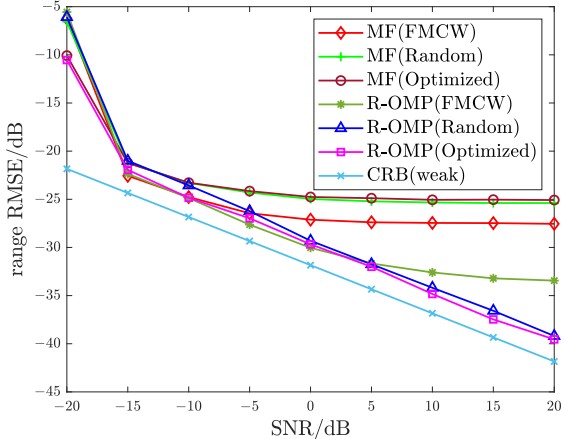

**Figure 13.** Range RMSE of strong scatterer.

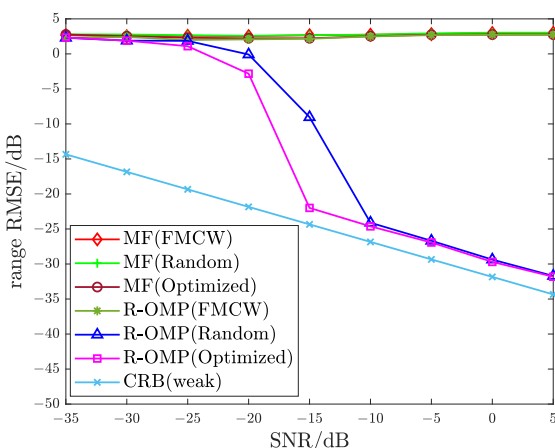

**Figure 14.** Range RMSE of weak scatterer.

## 7. Conclusions

In the paper, a waveform design approach and range estimation method were proposed in smart noise jamming scenes for high-resolution radar. The IPFC-FMCW waveform was utilized to solve the observations' periodically missing problem of the FMCW waveform in smart noise jamming scenarios. The simulated annealing algorithm was utilized to optimize the waveform parameters based on the range profile to obtain a better waveform performance after jamming suppression. Considering the missing observation caused by jamming suppression, the proposed range recovery algorithm is suitable for jamming scenarios. Experiment results showed that the proposed waveform and the range recovery method outperform the compared methods. For subsequent angle estimation, we perform the corresponding sum-difference angle measurements or array angle measurements, which will be discussed in future work.

**Author Contributions:** Conceptualization, R.X. and D.M.; methodology, R.X. and X.L.; software, R.X.; validation, R.X., L.W. and Y.L.; formal analysis, R.X.; investigation, R.X.; resources, X.L.; data curation, R.X.; writing—original draft preparation, R.X.; writing—review and editing, L.W.; visualization, D.M.; supervision, Y.L.; project administration, Y.L.; funding acquisition, Y.L. All authors have read and agreed to the published version of the manuscript.

**Funding:** This research received no external funding.

**Data Availability Statement:** Not applicable.

**Conflicts of Interest:** The authors declare no conflict of interest.

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
