# Peer review of "Intra-Pulse Frequency Coding Design for a High-Resolution Radar against Smart Noise Jamming"

_remotesensing, doi:10.3390/rs14205149_

Round 1

Reviewer 1 Report

see attached

Author Response

We thank the reviewer for the positive feedback and many important comments on the manuscript. We discuss each issue point-by-point in the attachment below.

Reviewer 2 Report

Summary:

This is a meaningful work.

Smart noise jamming forms active jamming by intercepting, modulating, and forwarding radar signals into the radar receiver, which seriously affects the radar range recovery performance.

This paper proposes a novel waveform design approach and an efficient range recovery method for high-resolution radar in the jamming scenario. Next, based on the designed waveform, this article derives the echo model after jamming suppression and propose a gridless compressed sensing (CS) method to recover the range of the targets.

The paper can be improved in the following aspects:

Detailed comments:

1. In Introduction, both of the methods you mentioned for transmitting large bandwidth signals are related to fast moving target detection, but this article seems to have nothing to do with fast moving target detection. I suggest author think it over.

2. In Section 2.1, can you make it clearly why jamming signals cause range recovery problem?

3. In Section 4.3, you said “choose a set of waveform parameters conforming to the constraint conditions as the initial values”, can you explain what constraint conditions are?

4. In Section 5, page 10, the discussion of Algorithm 1 Simulated Annealing is not suitable.

5. In Section 6.3, you said “we simulate the performance of scatterer recovery versus different radar SNR”, is SNR calculated according to “missing data” in Section 4? If not. I suggest author can add experiments to illustrate the effectiveness of the algorithm.

Author Response

(The authors gave the same response as above.)

Reviewer 3 Report

A waveform design approach and range estimation method were presented in smart noise jamming scenes for high-resolution radar in this paper. This paper is well organized and has a better performance improvement from the experiment results compared with traditional methods. Some advices are given as follows,

1.The signal transmission of the ?-th chip begins at zero while ,so it may not be correct for the equation in line 124.

2.In line 187, it should be “in a certain range” instead of “in the a certain range”. It is suggested to make a total check on the manuscript.

3. I would like to suggest to make some necessary comparison with different type of algorithms for designing frequency coding waveforms, and giving constrains on the design.

4. With the optimal compressive sensing recovery, the range reconstruction seems achieve improved performance. It is encouraged that investigation on the phase recovery precision would be useful to ensure the following sum-diff angle measurement.

Author Response

(The authors gave the same response as above.)

Reviewer 4 Report

In this work, the authors proposed a waveform design method an an efficient range recovery method, an intra-pulse frequency-coded frequency-modulated continuous waveform, for high resolution radar when a signal jamming mechanism is in effect. This research is of interest to the radar research community, especially for the high resolution radar research, where a fast range recovery is important, when a jamming signal is present.

However, a few issued required to addressed before the paper could be published.

1) From the article, it is not clear what is the novel method here. An elaborated description would help the readers.

2) The paper is too long. Few general descriptions, such as how the jamming signal works might be dropped off. A concise description with references would be sufficient.

3) It would be a good idea to include units of each parameters.

4) Figure 5 requires more discussion.

5) Figure 9 requires more discussion

6) On the page 7, line 5. The authors described, “By ignoring the influence of different antenna delays on the window function, ˜? (?) is..”

However, antenna delay would play a crucial role. Please elaborate why the antenna delay is ignored.

Author Response

(The authors gave the same response as above.)
